Basis of single-seed formation in chestnut: cytomorphological observations reveal ovule developmental patterns of Castanea henryi

Qiu Qi 1 2
Tian Xiaoming 3
Wu Guolong 1 2
Wu Juntao 1 2
Yuan Deyi 1 2
http://orcid.org/0000-0002-1633-0843 Fan Xiaoming 1 2 fan_xiaoming001@163.com
1 Key Laboratory of Cultivation and Protection for Non-Wood Forest Trees, Ministry of Education, Central South University of Forestry and Technology , Changsha, Hunan Province , China
2 Key Laboratory of Non-Wood Forest Products of State Forestry Administration, Central South University of Forestry and Technology , Changsha, Hunan , China
3 Hunan Botanical Garden , Changsha , China
Baskin Tobias
Electronic publication date: 2025 Jan 2
Publication date: 2025
Volume: 13
Electronic Location ID: e18711
Received 2024 Mar 20; Accepted 2024 Nov 24
Copyright: © 2025 Qiu et al.
Copyright year: 2025
Copyright holder: Qiu et al.
License: This is an open access article distributed under the terms of the Creative Commons Attribution License, which permits unrestricted use, distribution, reproduction and adaptation in any medium and for any purpose provided that it is properly attributed. For attribution, the original author(s), title, publication source (PeerJ) and either DOI or URL of the article must be cited.
License URL: https://creativecommons.org/licenses/by/4.0/

Keywords: Castanea henryi, Single seed, Ovule development, Embryo sac, Abortive processes

Funding: Natural Science Foundation of Hunan Province 2022JJ30997 Science and Technology Innovation Program for Graduate Students of Central South Forestry University of Science and Technology 2023CX02050 This work was supported by the Natural Science Foundation of Hunan Province (grant no. 2022JJ30997) and the Science and Technology Innovation Program for Graduate Students of Central South Forestry University of Science and Technology (2023CX02050). The funders had no role in study design, data collection and analysis, decision to publish, or preparation of the manuscript.

==============================
Background

Many plants, including those commonly found in the Fagaceae family, produce more flowers and ovules than mature fruits and seeds. In Castanea henryi, an ovary contains 16–24 ovules, but only one develops into a seed. The other ovules abort or otherwise fail to fully develop, but the reason for this is unknown. Such a strict reproductive screening mechanism is rare in plants.

Methods

In this study, controlled pollination scheme were adopted, and conventional paraffin embedding and semi-thin sectioning techniques, followed by microscopy, were used for cytological studies of ovule development in C. henryi.

Results

Pollination affected not only the process of ovule development, but also the proportion of ovules that formed mature embryo sacs. Approximately 53.53% of the ovules in the pollinated treatment developed normally, while only 16.55% of the ovules in the unpollinated treatment developed into mature embryo sacs with a seven-cell, eight-nucleated structure. Failure to form mature embryo sacs and the abnormal divisions of the zygote, respectively, were the reasons for the pre- and post-fertilization ovule failures. Our findings not only provide basic information on the reproductive biology and also information on seed production of C. henryi.

Introduction

During the development of angiosperms, many flowers, ovules, and seeds fail to develop into mature fruits and seeds. Factors such as the inherent characteristics of the species, number of flowers or ovules, processes of pollination and fertilization, and environmental conditions regulating plant growth are closely associated with this failure in development (Zhang et al., 2020; Severino, 2021). The period of ovule development is a crucial phase in the life cycle of plants as it determines reproductive output and offspring viability. In the early stages of the life cycle, viability selection, characterized by high numbers of ovules and a high rate of developmental failure, plays a pivotal role in shaping the genetic composition of plant populations. Importantly, this viability selection can occur at any stage of ovule development (Hufford & Hamrick, 2003). A certain degree of ovule failure is potentially a result of a viability selection process in which rigorous mate selection and selective fertilization are utilized to maximize reproductive success (Susko, 2006; Hasegawa, Suyama & Seiwa, 2009). However, seed failure has become a central problem in food production (Alqudah, Sharma & Börner, 2021). Moreover, ovule failure is a more effective mechanism than seed failure in reducing the costs of fruiting. As ovule abortion occurs earlier in the developmental sequence than seed abortion, the cost of fruiting is substantially reduced. This early-stage abortion allows plants to reallocate and increase the availability of conserved resources (Calviño, 2014).

The phenomenon of multiple ovules but single-seeded fruiting is common in the Fagaceae family (Boavida, Varela & Feijó, 1999; Fan et al., 2015). Castanea mollissima has 12–18 ovules in one ovary, but only one develops into a seed (Du et al., 2021). Similar phenomena also occur in C. crenata (Nakamura, 2003) and C. sativa (Viejo et al., 2010). As only one seed is present in an involucrum, there is usually one ovary per involucrum in C. henryi, which has been used as a model to research selective ovule development. This species is vital in targeting poverty alleviation in China’s mountainous regions, and its production has become the dominant industry in some areas (Zhang et al., 2016; Li et al., 2019b). The ovary of C. henryi contains 16–24 ovules with axile placentation. However, only one ovule usually maintains development and ultimately forms a seed, while the rest abort and become brown and withered (Fan et al., 2015; Qiu et al., 2023). The distribution of ovules within the ovary that mature into viable, fully developed seeds within the fruit is apparently at random. In a stringent selection process, mate choice and selective fertilization are employed to maximize reproductive success. Previous studies have shown delayed fertilization in C. henryi, and it takes 6 weeks to form a mature seven-cell eight-nuclear embryo sac structure (Fan et al., 2015). During those 6 weeks, approximately half of the ovules in the ovary are aborted due to abnormal development. Even if a mature embryo sac is formed, only four to five ovules are successfully fertilized. However, the steps that convert an ovary with several fertilized ovules to an ovary with only a single developing fruit are largely unknown.

In this study, we used microscopic sectioning techniques to reveal examined, at the cytological level, the ovule abortion before and after double fertilization and the characteristics during characterized the formation of a single seed. This research serves to provide baseline knowledge for future investigations of C. henryi or Fagaceae species reproduction.

Materials and Methods

Plant materials

The experimental site was located on the western campus of the Central South University of Forestry and Technology, Changsha, Hunan Province, China. The Chinese Chinquapin (Castanea henryi) cultivar Huali 4 (Variety No.: XiangS-SC-SH-010-2015) was used for observation, and the cultivar Huali 2 (Variety No.: XiangS-SC-SH-008-2015) was used as the pollen donor. The female inflorescences were collected between June and September 2021 for processing and preservation. The trees were approximately 3–4 m tall and grown for 8 years, and they exhibited normal blossoming and fruit-bearing capacities. Pollination and non-pollination treatments were applied to the flowers of the same tree. Three trees were treated, and about 120–150 flowers were sampled from each tree.

Pollination treatment

Male inflorescences of experimental materials were removed, and the female flowers which stigmas were just present were bagged to avoid the entry of exogenous pollen. A total of 400 female flowers were treated to ensure sufficient material for subsequent sampling. The anthers were peeled off from the staminate catkin with toothbrush, and were spread over paper sheets in a ‘pollen room’ (28 °C, 40% RH, 4,000 lx) for 4 h in order to loosen the pollen. Pollen was collected in dry covered 1.5 mL centrifuge tubes, and temporarily stored at 4 °C. On days 5 to 7 of flowering, when the angle between the stigma and the flower mid-axis was about 30° to 45°, pollination treatment was carried out on 200 female flowers in sunny and windless weather, and bags were put on immediately after pollination. The other 200 female flowers were not pollinated. In both treatments, the bags were removed 2 weeks after pollination when the stigmata of the female flowers turned yellow-brown. Samples were taken weekly after treatment. The two main developmental stages we investigated were the mature flower and the young fruit after fertilization.

Paraffin sectioning

The ovary was peeled off from the involucrum and fixed in an ethanol-acetic acid mixture (3:1, v/v), pumped, fixed in an acetic acid mixture for 6 h, and stored in 70% ethanola t 4 °C until needed. Using the conventional paraffin section method described by Fan et al. (2015), samples were sectioned 8 µm thickness with a Leica rotary microtome (RM2235; Leica, Heidelberg, Germany). The dewaxed sections were stained with safranin and Fast Green, and the slides were sealed with neutral balsam. Sections were examined, photographed, and imaged using an optical microscope (BX-53; Olympus, Tokyo, Japan).

Semi-thin sectioning

The samples were fixed in 2.5% glutaraldehyde in 1× phosphate-buffered saline. The samples were then dehydrated with an ethanol gradient series and embedded in EMbed 812 resin. The embedded samples were cut to 2 µm with a glass knife in an ultramicrotome (UC7, Leica) and stained with a mixture of 0.5% safranin and 1% methyl violet (1:1, v/v) (Tolivia, Navarro & Tolivia, 1994). Stained sections were viewed and photographed using the BX-51 optical microscope.

Results

Ovaries and nuts

C. henryi is a monoecious species, bearing two types of catkins: the staminate catkins and bisexual catkins with female flowers that usually occur singly (Fig. 1A). In this species, a ring of green acuminate bracts covers the female flowers, and placental hairs or trichomes in the placenta protect the ovary. The involucrum progressively hardens into a burr when the female flower differentiates into the fruit (Fig. 1B). The pistils of C. henryi comprise the stigma, style, and a large ovary (Fig. 1C). When mature, the burr naturally cracks open, revealing the nut inside. Mature C. henryi nuts are conical in shape and their shells are smooth, hard, reddish brown, or bright brown (Fig. 1D).

Figure 1 Inflorescence, pistil and realistic pictures of single seed fruiting of C. henryi.

(A) Bisexual inflorescence. (B) Longitudinal section of female flower. (C) Pistil. (D) Fruit dehiscence state. inv, involucrum; sty, style; sti, stigma; tr, trichomes; ova, ovary.

Development of the ovule

In the early stages, there was no evident difference in ovule size within the same ovary (Fig. 2A). At 5–6 weeks after pollination, ovules continued to develop with enlargement of the ovary (Figs. 2B , 2C). At week 7 after pollination, the volume of one ovule was visibly larger than that of other ovules, and it was full and rounded (Fig. 2D); this ovule has already been fertilized. With the continuous expansion of the ovary, the fertilized ovule continued to develop, showing a milky white color, and the volume becoming much larger than that of the other ovules. Conversely, the other ovules gradually browned (Figs. 2E, 2F). In flowers that were bagged and not pollinated did not have a single prominent ovules, indicating fertilization. All ovules in these ovaries gradually browned, atrophied, and finally formed empty structures (Figs. 2G–2I).

Figure 2 Developmental process of C. henryi ovule.

(A–C) Represent the ovary developed at the 4th, 5th and 6th week after pollination, respectively. (D–F) Represent the pollinated ovary developed at the 7th, 8th and 9th week after pollination, respectively. (G–I) Represent the non-pollinated ovary that abnormally developed at the 7th, 8th and 9th week after pollination, respectively. The red arrow indicates the fertile ovule. All of the above images are of C. henryi viewed under a stereomicroscope after stripping the ovary wall.

Double fertilization

We observed that the ovules of C. henryi were enveloped by inner and outer integuments to form a micropyle. The development type of embryo sac was typically polygonum-type with double integuments and thick nuclear tissue. C. henryi had delayed fertilization and immature ovules during blooming. Approximately 6 weeks after pollination, they developed into mature embryo sacs with typical seven-cell and eight-nuclear structures (Fig. 3A). The egg apparatus comprised an egg cell and two synergids near the micropyle, and the nucleus of the egg cell was inclined toward the chalazal end. The synergid cell was pear-shaped, and its nucleus was inclined toward the micropyle end. The central cell was in the middle of the embryonic sac (Fig. 3A). The antipodal cells were close to the chalazal end and degenerated shortly after the embryo sac matured.

Figure 3 Egg apparatus of mature embryo sac and process of double fertilization of C. henryi.

(A) Eight-nucleate embryo sac. (B) Pollen tube entering embryo sac and releasing two sperm cells. (C) Sperm cell approaching egg with help of antipodal cell. (D, E) Sperm nucleus gradually moving closer to egg nucleus. (F, G) Sperm cells and egg cells gradually fusing and male chromatin dispersing into egg nucleus and male nucleoli appeared. (H, I) Zygote. (J–l) Sperm nucleus moving toward polar nucleus. (M) Sperm nucleus attached to polar nucleus and gradually fusing. (N, O) Male chromatin dispersing into polar nucleus. (P) Male nucleus appearing in polar nucleus. (Q) Primary endosperm nucleus forming. OI, outer integument; II, inner integument; NU, nucellus tissue; EN, egg nucleus; SC, synergid cell; PN, polar nucleus; SN, sperm nucleus; EC, egg cell; MI, micropylar side; MN, male nucleus; Zy, zygote; PEN, primary endosperm nucleus. (A)–(Q) are paraffin sections double-stained with fuchsin-solid green.

Two sperm cells appeared in the embryo sac (Figs. 3B, 3C). The sperm nucleus was seen in some images in close proximity to the egg nucleus (Figs. 3D, 3E), and in other images the sperm nuclei appeared attached to the egg nuclear membrane and undergoing the process of fusion (Fig. 3F). In images where karyogamy appeared to be taking place, the chromatin of the male nucleus was diffuse, and a small nucleolus appeared (Fig. 3G). In some images, the egg sac appeared to contain a zygote (Figs. 3H, 3I). In some sections, a sperm nucleus was seen near the polar nucleus in the central cell (Figs. 3J–3L ), and in some images, the sperm nucleus and the nuclear membrane of the polar nucleus were apposed and appeared to be fusing (Fig. 3M). The chromatin in the sperm nucleus was either displaced from one side to the polar nucleus or spread throughout (Figs. 3N, 3O). A nucleolus was also observed (Fig. 3P). The two nuclei fused to form the primary endosperm nucleus, which was located near the middle of the embryo sac (Fig. 3Q). The contact between the sperm and secondary nuclei preceded the contact between the sperm and egg nuclei.

Embryonic development

The egg cells formed zygotes after successful double fertilization. After the dormancy period and multiple divisions, the globular embryo gradually formed at week 8 after pollination (Figs. 4A–4E). The globular embryos were relatively regular (Figs. 4D–4F). The cotyledon primordia then developed on both sides of its top, forming a heart-shaped embryo (Figs. 4G, 4H). The final mature embryo and integument grew synchronously and filled the entire ovary. The nuts finally matured at week 20 after pollination.

Figure 4 Embryonic development of C. henryi.

(A, B) Zygotic first division. (C) Zygotic second division. (D–F) Globular embryo. (G, H) Gradual formation of heart-shaped embryo. (I) Mature seed kernel. EC, endosperm cell; Ke, kernel. Scale bars = 10 μm. (A)–(H) are paraffin sections double-stained with fuchsin-solid green.

The embryonic development of C. henryi did not include the formation of an obvious radicle. The development type of the endosperm was nuclear. The endosperm first formed in the middle of the ovule and approached the end of the micropyle at the globular embryo stage. At the heart-shaped embryo stage, the endosperm cells began to degrade their cell walls, and became distributed in the embryo sac in a free state (Figs. 4G, 4H). The endosperm presumably provided nutrients to the embryo during development, there was no endosperm when the kernel was mature (Fig. 4I).

Ovule abortion before double fertilization

Abortions in plant seed formation are complex and diverse. Abortion may occur at any stage, ranging from the archesporial cell to embryonic development and maturation. In C. henryi, some ovules prior to fertilization had embryo sacs that appeared to be abnormal even tough the integument appeared to be developing normally. In a single ovary at the mature embryo sac stage, approximately half of the ovules exhibited abnormal embryo sac development (Table 1). The abnormalities can be categorized into three types: the embryo sac cavity was elongated and narrow (Figs. 5A, 5B); there was a normal embryo sac cavity, but the structure of the egg apparatus structure was missing or incomplete (Fig. 5C); and embryo sac degeneration occurred. Embryo sac degeneration was manifested as follows: the embryo sac sometimes developed to the four-nucleate stage (Fig. 5D); sometimes, at the late stage of embryo sac development, the central cell (Fig. 5E), antipodal cell (Fig. 5F), and egg apparatus could appear to be degenerated (Figs. 5G, 5H), and finally, sometimes abortive cells with abnormal embryo sac development showed degeneration of embryo sac, shrinkage, and deformation of nucellus tissue cells, as well as degeneration and disintegration of nucellus tissue (Fig. 5I).

Table 1 Proportion of normal ovule development in single ovary in two treatments.

Weeks after pollination	Developmental stage	Proportion of normal development ovule	
Pollinated	Unpollinated	p	
1	Archesporium, megasporocyte	100%	100%		
2–3	Megasporocyte, meiosis, functional megaspore	100%	100%		
4–5	Uninucleate embryos sac, two-to eight-nucleate ES	67.67 ± 1.38%	42.24 ± 2.34%	<0.001***	
6	Mature ES	53.53 ± 2.40%	16.55 ± 1.44%	<0.001***	
7	Fertilization	20.36 ± 1.30%	0%	<0.001***	
Note:

Data are means (±S.E) for 20 replicates (N = 20). Statistical comparisons were performed using the t test. T-test for all variables, ***p < 0.001.

Figure 5 Ovule abortion before fertilization.

(A, B) Failure to form embryo sac cavity leads to ovule abortion. (C) Abnormal egg apparatus development. (D) Degenerated four-nucleate embryo sac (arrow). (E) Degenerated central cells in aborted ovules. (F) Degenerated antipodal cells within the abortive ovule. (G, H) Egg apparatus degeneration. (I) Germ cell degeneration and nucellus tissue atrophy. AEA, abnormal egg apparatus; DAC, degenerate antipodal cells; DEA, degenerate egg apparatus. Scale bars: 100 μm (A), 50 μm (B, C), 10 μm (D–I). (A)–(C) are paraffin sections double-stained with saffron-solid green; (D–I) are semi-thin sections stained with saffron-methyl violet mix.

To investigate the impact of pollination on ovule development, we subjected female chestnut flowers to two treatments: pollination and non-pollination. Compared to pollinated ovules, non-pollinated ovules exhibited relatively delayed development, with a reduction in the number of ovules appearing with normal development at each stage. Approximately 53.53% of ovules in pollinated ovaries developed into mature embryo sacs with a seven-cell eight-nucleus structure (Table 1). Without pollination, only about 16.55% of ovules in a single ovary had a mature embryo sac structure (Table 1). Pollination partially influences the developmental outcome of the embryo sac.

Abortion of ovules after double fertilization

Not all ovules formed seeds after successful fertilization. In the semi-thin sections, approximately 16% of the ovules appeared to show post-fertilization abortion, which can be classified into three types: first, a normal proembryo structure could not be formed after fertilization, no normal zygote appeared, and only a clump of flocculent tissue was present (Figs. 6A–6C); second, the zygote formed, but it did not split (Figs. 6D–6F); and thirds, the zygote was formed, and the development of the zygote stagnated after the first normal division to form the second proembryo (Figs. 6G, 6H). If the ovule formed a zygote, then its endosperm developed normally from the primary endosperm nucleus to the free endosperm nucleus (Figs. 6I, 6J). Moreover, after the formation of the primary endosperm nucleus followed by a very short dormancy period multiple free endosperm nuclei were formed.

Figure 6 Ovule abortion after fertilization.

(A–C) Fertilized ovule does not form a normal proembryo structure. (D–F) Zygote formed after fertilization but did not divide. (G, H) Zygote divided to form a diploid embryo but did not continue to divide. (I, J) Normal division of primary endosperm nucleus to form free endosperm nucleus. PEN, primary endosperm nucleus; FEN, free endosperm nucleus; DE, diploid embryo; Zy, zygote. (A)–(J) are semi-thin sections stained with a mixture of senna-methyl violet.

Discussion

The extra production of ovules within flowers is commonly observed in various plant species (Sakai & Kojima, 2009). Each ovule has the potential for fertilization and can develop into a viable embryo and even mature seed (Yu, Jiang & Lin, 2022). However, in C. henryi, only one ovule usually develops into seeds; the reason for this remains unclear. Therefore, we investigated the anatomical characteristics of abortive ovules. C. henryi ovule abortion occurs during three stages of embryo sac development: before double fertilization, during double fertilization, and during zygote division. The single-seed formation in chestnuts results from a stepwise screening mechanism.

The embryo sac has a crucial role in the key process of sexual reproduction in plants. Several factors, including resource allocation (Lee & Bazzaz, 1982), positional effects (Silveira & Fuzessy, 2015), and deposition patterns of callose in ovules (Calviño & García, 2009), can impede embryo sac development. However, anatomical observations in chestnut ovules suggest that positional effects do not influence embryo sac abortion. In this study, anatomical observations of pollinated and non-pollinated ovules revealed that pollination affected embryo sac development. Similar phenomena have been observed in various plant species, including Orchidaceae (Mayer et al., 2021), in which pollination plays a crucial role in triggering or regulating embryo sac development and ovule maturation, ultimately enabling fertilization (Sakai, 2007). Similarly, in Ginkgo biloba, pollination triggers the development of the ovule integument (D’Apice et al., 2021).

On the premise that artificial pollination ensures successful fertilization, we found that the main reason for ovule abortion in chestnut after double fertilization was abnormal zygote development. Similar phenomena have also been observed in Hanfu apples and tetraploid black locusts (Yang et al., 2014). In the latter, the abortion rate at the zygotic stage is as high as 50% (Jiang et al., 2011). Selective abortion in plants is mostly achieved through programmed cell death, which may lead to ovule development deformity, nuclear tissue degeneration, embryo sac abnormality, zygote stagnation, or developmental interruption (Hauser et al., 2006; Wang et al., 2021). The process of nuclear tissue apoptosis in Ginkgo biloba has been verified as programmed cell death (Li et al., 2019a). In the following research, programmed cell death may be considered in the study of aborted ovules in C. henryi. Exploration of the molecular mechanisms underlying this process is a promising avenue for future study.

In summary, chestnut undergoes layer-by-layer elimination and selects presumably the highest quality ovules to form seeds for successful reproduction. Reducing ovule development may serve as a resource allocation strategy, especially considering the exceptionally high number of ovules produced in chestnut ovaries. This survival strategy is important for resource management within the chestnut ovary.

Supplemental Information

Supplemental Information 1 Raw images for Figure 1.

Supplemental Information 2 Raw images for Figure 2.

Supplemental Information 3 Raw images for Figure 3.

Supplemental Information 4 Raw images for Figure 4.

Supplemental Information 5 Raw images for Figure 5.

Supplemental Information 6 Raw images for Figure 6 (part 1).

Supplemental Information 7 Raw images for Figure 6 (part 2).

Supplemental Information 8 Statistical raw data on the normal development rate of ovules in single ovary at 6 weeks after pollination of two treatments in Castanea henryi.

Additional Information and Declarations

Competing Interests

Author Contributions

Data Availability

The authors declare that they have no competing interests.

Qi Qiu conceived and designed the experiments, performed the experiments, analyzed the data, prepared figures and/or tables, authored or reviewed drafts of the article, and approved the final draft.

Xiaoming Tian conceived and designed the experiments, analyzed the data, prepared figures and/or tables, and approved the final draft.

Guolong Wu analyzed the data, prepared figures and/or tables, authored or reviewed drafts of the article, and approved the final draft.

Juntao Wu performed the experiments, analyzed the data, prepared figures and/or tables, and approved the final draft.

Deyi Yuan conceived and designed the experiments, performed the experiments, authored or reviewed drafts of the article, and approved the final draft.

Xiaoming Fan conceived and designed the experiments, authored or reviewed drafts of the article, and approved the final draft.

The following information was supplied regarding data availability:

The raw measurements are available in the Supplemental File.

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
