# Peer review of "Basis of single-seed formation in chestnut: cytomorphological observations reveal ovule developmental patterns of Castanea henryi"

_PeerJ, doi:10.7717/peerj.18711_

## Round 0.1 · original submission · Major Revisions

Thank you for submitting your interesting study to Peer J. I apologize for the time that has elapsed so far in review. I hope at least you find the reviewers’ and my comments helpful.

All three reviewers found interesting material in your submission but all of them identified notable problems. I read your paper carefully and agree. Synthesizing reviewer concerns, I identify three broad themes. Note that in your resubmission, I expect you to submit a letter where you reply to all of the points raised in review. If there are those you disagree with, that’s fine, just explain why.

First, more information about the methods is needed. You do not describe how many samples were investigated. For each method (figure, table) you need to state carefully how many ovules were examined. You need to describe the pollination treatment much more carefully. At some places in the text you refer to manual pollination but others to natural “free” pollination. Did you do anatomy on both? Likewise, for non-pollinated material, apparently you placed a bag of some kind over the flower. You need to describe how all of this was done. I wonder if you have a control for the bagging: for example, were hand pollinated flowers also bagged? Bagging itself could be influential. You need to describe the scanning electron microscopy. How were the flowers handled to make Figure 2?

Second, the paper needs a better structure. One approach would be to present successful development first, starting from the formation of the megagametophyte through to the mature embryo. With that established, then show deviations from the normal. I think this would make your results easier to follow.

Third, is there really ovule abortion post fertilization? In Figure 2, the ovary clearly has a single dominant ovule on week 7 after pollination, which is when you claim fertilization happens. So for the ovary in Fig. 2, all of the abortion is evidently pre-fertilization (or at the same time as). How many of this type of observation did you make? And in Figure 5, which purports to show fertilized ovules, how do you know the structures being called “zygote” are actually zygotes rather than some unusual cell structures following on from the abortion of a mature egg sac?

It is important to distinguish days after pollination, which you can easily monitor, from days after fertilization, which you can only infer from careful observations.

Now, I would like to make a few points of my own.

When you present figures that you believe represent abnormal situations, be careful about what you are actually observing. I mentioned above the ambiguity about whether you are actually seeing fertilization or looking at samples that are staged as being fertilized (but not demonstrated as such). In other cases, you describe some of the figure panels in highly specific terms but it is not always clear from the actual images how you can be so specific. It is fine to refer to “atypical anatomical [or cytological] arrangement” without over-interpreting.

Be sure to refer to all figures. In the current submission, you have a Figure 7 and 8 but these are never referred to in the text. Likewise, you refer to supplementary figures 1 and 2 but these were not uploaded.


Although reviewer 3 felt that you have a good introduction and state the scientific question being asked, I disagree in part. In the two main paragraphs of your introduction, you certainly do a good job in addressing general issues surrounding ovule abortion. But then, specific issues for your paper are jammed into the last paragraph containing only four lines. The main part of your introduction might just as well be introducing an ecological or evolutionary study on ovule abortion. Instead, you need to introduce the reader to the specific question(s) that you are asking. I think you are asking at which stages do ovules abort in chestnut and possibly with what kind of cellular appearance. Presumably this has been studied in other species; maybe in some, all of the abortion happens before fertilization? Or all of it happens after? Maybe some kinds of structures are recognized in other species that undergo ovule abortion. These kinds of possibilities need to be introduced and relevant literature cited.

Figures 1 and 2 are a helpful attempt to overview fruit development in chestnut. But Fig 1 e - g, although interesting, are not helpful for the issues at hand. By contrast, I don’t understand the relation between Figure 1 c and b. Is c a later stage? Were the hairs and so on removed? Fig. 1d is almost impossible to see; the SEM seems dark and not clear and the many lines drawn over simply obscure the structure. Perhaps an image taken on a dissecting scope would be better? Note that it would probably better to have a second panel that is only diagram next to this image, rather than superimposing all of the annotation.

As one of the reviewers noted, your description of the inflorescence, flower, and fruit (the first section of the results) is quite difficult to follow. The terminology is confusing and I think possibly not entirely standardized. What might help here would be a general diagrammatic figure that would introduce the reader to the relevant stages and structures. And such a figure could establish the terms you are using. That way, even a term is not standard, at least readers will know what you mean.

As one of the reviewers did, I found the SEM images in Fig. 3 to be unconvincing. The samples look poorly prepared and I think the surface morphology is not particularly informative. Perhaps using a dissecting scope would be more appropriate? Or this figure could simply be deleted.

The images in figure 2 imply that only one ovule enlarges. Is this always true? If so that might indicate some kind of signal that tells the fruit: “OK, we just have the one good one, grow!” This seems interesting to follow up.

I noticed that you used an editing service. Your English is grammatical, which is important and good. But from a science (or content) standpoint, the language is frequently off. Reviewers call out some of these places. I will add a few more here. Line 175: “All ovules can not form seeds after successful fertilization.” Do you understand the difference between “can not” and “do not”? The latter (do not) is a statement of fact; whereas, the latter is a statement about potential. I don’t see how you can claim that they “can not” without a great deal more evidence. Line 189: “The zygote has the advantage…” Here advantage is not the correct word. I don’t know what you mean. Line 196: “karyotyped” means to count the chromosome number and characterize their lengths. That is not what you mean here. Line 203: The word “excessive” is a matter of judgement. Someone who drinks and gambles and parties all night can be accused of ‘excessive’ behavior. If you introduced a transgene that caused a plant to make dozens of additional ovules, those could be called excessive. But the ovule production in chestnut is presumably adaptive. Use a word like “unused” or “extra” that carries no judgment.

The above comments on scientific language make an important point about commercial editing services: Most of these can deal only with English grammar. This misses the level of scientific language. If at all possible, ask a colleague who is a native English speaker and works on plants to read and comment.

In the figure legends, please provide detail about whether the image is a plastic section or a paraffin section.

Reviewer 1 ·

Basic reporting

The paper by Qiu and collaborators aimed to answer why the chestnut forms a single-seeded fruit by analyzing the ovules development using light and scanning electron microscopy. Although the manuscript contains some interesting information, the authors need to go further in terms of description, interpretation, and discussion of the results.
The English language should be improved to ensure that an international audience can clearly understand the text. Some examples where the language could be improved include lines 181-185 – the current phrasing makes comprehension very difficult. I suggest that an authors’ colleague who is proficient in English and familiar with the subject matter review the manuscript, or the authors contact a professional editing service.
Other problems of the manuscript are listed below:
-The abstract does not clearly present the research question, the main results and conclusions.
-The Introduction contains sentences with incorrect information, for example:
“In the developmental process of both angiosperms and gymnosperms, a considerable number of flowers, ovules, and seeds fail to develop into mature fruits and seeds and perish.” (line 38-39) – Gymnosperms do not have flowers and fruits.
“The ovary of C. henryi contains 16-24 ovules with axial placentation.” (lines 65-66) – The placentation in this species is axile, not axial.
-The last statement of the Introduction “This research lays the foundation for solving the problem of empty burrs in C. henryi and provides a basis for improving the industrialized production of this species.” (lines 74-77) is not supported by the results.

Experimental design

The methods were not described with sufficient detail.
Reproductive tests were mentioned in the results (lines 124-125; 131-133), but were not described in the methodology. How were these tests carried out?

Validity of the findings

-Some parts of the results are very confusing and need to be rewritten (for example, lines 108-111). The authors should bring the description of the results in accordance with terminology accepted in the plant morphology and embryology books. There are some terms used in the text that are not appropriate, for example, germ cell (lines 155, 159). In addition, the description of the results contains conceptual errors, for example: -figure 1G shows the fruits (nuts), not the seeds as described in the legend; - placental hairs are trichomes that occur in the placenta (inside the ovary), not in the involucrum as described in line 112 and indicated in the fig. 1B.
-In relation to the figures, some of them are not illustrative. In the fig. 3 the samples showed in SEM micrographs are not well preserved (the cells are collapsed), thus the images do not allow for a proper comparison of the ovule surface between fertile and aborted ovules (and I do not think that such comparison is relevant in the context of the work). In addition, the images showing the zygote (figs. 5A-F), the two-celled embryo (Fig. 5G-H), and the primary endosperm nucleus (fig. 5I) are not convincing. What is the evidence that such ovules were fertilized?

Additional comments

Considering the comments given above, I cannot recommend this manuscript for publication in PeerJ. The text contains several inconsistencies and requires extensive rewriting to show its main findings and contributions.

Reviewer 2 ·

Basic reporting

1. The introduction need more in detail. Line 56-57, What is the cause of these phenomenon (ovule abortion) in the references (Boavida et al., 1999; Fan et al., 2015)?
2. The English language should be improved to ensure that audience can understand your work clearly.
3. It is suggested to compare the developmental structure of fertile ovule and sterile ovule within the same ovary of different developmental stages. Rather than, the development of sterile ovule was described first in the manuscript, followed by the development of fertile ovule.
4.The format of the figure and figure legend is confused. In Fig.2, The label of the pictures should be uniform and in lower case letters, and is it an error in the serial number of the fig d-f? In Fig3, the ovule after fertilization should be labeled as a young seed. The abbreviations in fig.8 should be arranged in order a-z. The format is consistent with that of the previous figure legends. MN repeated.

Experimental design

How many developmental stages were investigated in this study? Range from buds, mature flowers to young fruits after fertilization.The developmental stages of plant materials is not clear enough.

Validity of the findings

How many female flowers (or ovary, or ovule) were counted in this study, and what proportion of these three types of ovule abortion occurs respectively? As the authors stated, “In the process of single-seedformation in C. henryi, ovule abortion primarily occurs in three stages: pre-fertilization embryo sac development, double fertilization, and post-fertilization zygote division.”

Additional comments

The family name of Orchidaceae should not be written in italicized(Line 221).

Reviewer 3 ·

Basic reporting

• Clear, professional, English language is used throughout, except for a few minor points (given in general comments).
• Literature is well referenced and relevant. The into and background generally give sufficient context. I appreciated that the background included both a broad perspective about the research question in the context of angiosperm reproduction, but also focused information about the research question in a genus-specific context. I think readers would benefit from additional introduction about the timeline of ovule/seed development in relation to pollination (specifically, that ovule maturation does not occur until well after the time of pollination).
• Structure conforms to discipline norms.
• Figures are relevant, generally sufficient quality. Raw data is supplied (additional explanation/labeling of Supplementary table 1 is necessary). I think it would be useful to readers to have a figure (or supplementary figure) with normal/healthy ovules development in C. henryi (as a complementary figure to Figure 4, which shows abnormal / aborted ovule development). Such a figure is not completely necessary, however, and a healthy, mature embryo sac is already provided in figure 8.
• Manuscript is self-contained and represents a reasonable unit of publication.

Experimental design

• Original Primary research is within scope of the journal.
• Research question is well defined and well justified. It is stated how the research fills the knowledge gap.
• Investigation is generally performed to a sufficient standard (additional details about experimental design is required, see general comments).
• Technical methods are described in sufficient detail. Additional details about experimental design is required, see general comments.

Validity of the findings

• Underlying data appears to be provided and properly controlled. Additional details are required about experimental design, which will determine the appropriateness of the statistical analysis provided.
• Conclusions are well stated and linked to research questions, and limited to the supporting results.

Additional comments

Summary: The manuscript “Why does chestnut for a single seed per ovary? Cytomorphological observations reveal developmental patterns of Castenea henryi” presents observations on ovules and seeds from pollinated and un-pollinated fruits of Castanea henryi, which is an economically important crop in regions of China with potential for expansion of its economic impact. The authors determine that pollination impacts the development of fertile ovules, and that ovule or seed abortion can occur at multiple stages (pre-fertilization, fertilization, and post-fertilization). Furthermore, the authors determine the frequency of different types of developmental failures at those three stages.

Line 34: What was the cellular mechanism that was elucidated?
Line 45: should “fertilized” be deleted? As it is discussed and shown I the results that pre-fertilization developmental failure can contribute to the number of viable seeds.
Line 80-85 (and supplementary table 1): please provide more detail on the experimental design / sampling design. For example, the columns in supplementary table 1 should be labeled. Were multiple trees sampled (how many)? Were multiple flowers per tree sampled (related to confusion about lines 109-116, see below)? Were pollinated and non-pollinated treatments done on flowers from the same tree?
Line 101-104: I can see that t-test results are reported as part of table 1, but I do not understand the notation. What pair-wise comarisions were preformed? The table seems to indicate that there were no significant differences between the pollinated and un-pollinated treatments at all stages. Please explain the t-tests in the results text.
Line 109-116: I am confused by the use of “flower” vs “inflorescence” in the section. For example, line 109 states that “The male flower is a unisexual inflorescence composed of only male flowers, which are mixed inflorescences”; inflorescences are comprised of flowers, not the other way around. Similarly, line 114-115, “female flowers… have just one female flower cluster in the involucrum” needs to be corrected.
Lines 124-134: when “fertile ovule” is used, do you mean “fertilized ovule”?
Line 147: “Female abortions” – given that post-fertilization developmental failures are included, not all seed abortions are necessarily under maternal control (could be due to embryo and/or endosperm developmental failure that is independent of maternal influence).
Lines 166-167: ovules do not develop into mature embryo sacs. Rather, ovules can contain mature embryo sacs.
Line 172: what does “material” in “material embryo sacs mean?
Lines 183-185: do you mean that multiple free endosperm nuclei were formed after the dormancy of the primary endosperm nucleus?
Line 225: is it known whether artificial pollination does ensure successful fertilization in this system?
Line 234: how do you know that the observed cell degeneration was programmed cell death?
Line 236-237: what does “layer-by-layer” and “batch-by-layer” mean? Is this related to the experimental design?
Line 244: I don’t think “monocyte” is the right word here. Should be “single seed”?

Figure 2: The labels on the last three panels (bottom row) need to be fixed (G, H, and I, instead of d, e, and f).
Figure 3: please re-label the figure panels to be A-M, rather than a, ai, aii, b, bii…
Figure 4: it would be useful to have documentation of similar stages of normal/healthy ovules development
Table 1: What is the “n” of n = 3? Is that individual inflorescences or flowers?
Figure 7: What is a “structural deficit” (as compared to the other categories)?

---

## Round 0.2 · Major Revisions

This paper has been much improved. Both of the reviewers are generally supportive and I agree. But both reviewers found issues, about which I also agree. I read the paper carefully and I need to point out that a few further issues must be clarified.

One important issue are the data shown in Figure 5. On its face, this figure makes no sense. The figure shows data for three different times (0-6 weeks, 7 weeks, and 7 - 9 weeks), giving the percentage of various kinds of ovules within a single ovary. But most of the percentages are exactly the same for the three times. That cannot be real data (the percentages might be similar but not exactly the same) because these are destructive samples. Something is wrong here. Please revise completely to present what in fact you measured. The revision must include information about how many ovaries were sampled at each stage and the variability among the ovaries in the given percentages. That is, present mean ± standard deviation (or some other relevant indication of the variation).

Also problematic, the data in figure 5 partially overlap that of Table 1, maybe. In general, it is not clear whether the images and data for pollinated material come from bagged or open pollination. This should be made clear throughout.

Another issue is the staging of observations and figures. For example, the way figure 7 is described, the text reads as though the results are for ovules that have been fertilized. But I don’t see how it is possible to know that. Instead, I think that this material was sectioned at week 9 or 10. In other words, based on timing it is “after fertilization”. But the ovule itself might never have been fertilized. To avoid this ambiguity, please use the time of sampling to stage the data (sections, observations, etc).

There are also a large number of small problems. Some of these are with English but others relate to the content. I have marked up the manuscript extensively, with edits and comments. (A few of these overlap with points raised by the reviewers). I am not going to repeat them here. But please download my marked up version and look at all of the suggested changes and comments. If you have specific questions you are welcome to email me off line ([email protected]). I’ll be happy to help.

Reviewer 3 ·

Basic reporting

Basic Reporting: Basic reporting criteria are as before, with additional/correct background information. Reorganization of figures and presentation of “normal” development has increased the readability of the manuscript.

Experimental design

Experimental Design: Experimental design are as before, with added clarification of experimental design. Some additional clarifications are needed (see comments below).

Validity of the findings

Validity of Findings: Validity of finding criteria are as before, with revision of wording in the conclusions to improve the accuracy of the conclusions. Minor edits in the relevant section of the abstract is needed to reflect changes made in the main text conclusions.

Additional comments

Overview: The authors put considerable towards addressing reviewer and editor comments, including creating new figures that establish normal embryo sac and offspring development (which acts as a base of comparison for interpreting abnormal/abortive developmental trajectories). However, a few issues remain (see comments below), notable related to details of sample collection and data recording.

Additional Comments:
Line 35 : As originally commented, what was the cellular mechanism that was elucidated?
Line 91: manuscript text notes 120-150 flowers were sampled from each tree, but the author response to comments notes 100 flowers were sampled from each tree. In addition, were 100 (or 120-150) flowers per tree from each treatment (pollinated and unpollunated) sampled, or were 100 (or 120-150) flowers sampled per tree in total? In line 99, it is stated that there were 200 flowers per treatment; is that 200 per treatment per tree, or 200 total? Either way it’s inconsistent with the previous description.
Line93-94: please revise “stigmata are present but do not have the pollination ability of the parent tree for artificial de-male treatment.”, as it is unclear what this means. I also will guess that “stigmata” should be “stigma”.
Line 150: “synergism” should be “synergid”
Lines 154-167: I’m impressed with the numerous details/stages of double fertilization that the authors now present. Some figures do not readily show the described features (see following comments) and higher images resolution would help with image interpretation.
Line 155: image resolution is not high enough to determine if the sperm cells are present in the image.
Line 158: sperm nucleus not obvious in Fig 3e (is blocked by arrowhead?)
Line 259: What is the evidence for the chestnut mother plant selecting highest quality ovules? It is shown that a subset of ovules are fertilized and that a subset of seeds develop to maturity, but the relative quality of the ovules was not assessed.
Supplementary Table 1: labels have been added to most columns, but not all. Even with some of the labeled columns, the meaning of the values is unclear (example, the column titled “normal development in pollinated” has values 1-5. What do 1-5 mean? What are columns B and I? Why is column H blank for rows 47-61 (but column I is not blank)? What are the values in the “Avg” and “STDEV” column averages and standard deviations of? What does each row represent (such as, an individual sampled flower/fruit, branch, or tree)?

·

Basic reporting

The manuscript has been improved after previous revisions. However, some aspects still need correction and rewriting. The term “involucrum” is used in the introduction (line 60), which I am not familiar with. Is this term commonly used for the family or species? What would this structure be? (which part of the plant)
I was particularly confused by the following: "it also has three female flower clusters enveloped by an involucrum". Is each flower enveloped by an involucrum? Or is the group of three flowers enveloped by a single involucrum?
Please, provide some information about this structure in the text and clarify these doubts.

Experimental design

- Authors could include "(no more pollination occurs)" or something similar after "female flowers turned yellow-brown" (line 102)

- Line 109: replace "Leica rotary slicer" by "Leica rotary microtome"

- In the methodology, the topic "Materials treatment" should be replaced by "Pollination treatments" or something similar. Furthermore, although it was clear that 200 flowers were used in pollination tests, it was not mentioned which tests were performed (only the Results mention the tests: self, outcrossing, or natural pollination - line 132). Rewrite this topic, including how the pollination tests were done.

Validity of the findings

- Considering the sentence ""... and placental hairs or trichomes in the placenta protect the ovary" (lines 125-126) and observing Figure 1B, these trichomes do not originate from the placenta of the ovary, as they occur externally. Perhaps these trichomes are from the external surface of the pistil or internal surface of the involucrum. Confirm the origin of the trichomes and make the necessary corrections.

- Lines 128-129: "When mature, the thorns naturally crack, revealing the nut inside". It seems to me that these thorns are part of the involucrum, so it seems more correct to say that the thorny involucrum naturally cracks, revealing the nut inside.

- In the topic "Development of the ovule" of the Results, the different pollination tests (self, outcrossing, or natural pollination) did not change the final result (development of only one seed). However, the pollination experiments were not well described, which compromises the validation of the data.

- Line 139: Figure 2e-f is shown. What about Figure 2d?

- The phrase "with obvious volume expansion at week seven after pollination" (lines 140-141) can be removed, as it refers to the experiment without pollination.

- Line 150: Is "synergism" correct?

- Line 157: synergistic cells?

- Lines 181-183: "Even if the zygote divided normally, abortion may occur at all stages of embryonic development in fertile ovules, leading to the formation of empty buds". So abortion occurs in pollinated flowers. But in which of the pollination treatments (self, outcrossing, or natural pollination) does this occur? This needs to be included and discussed.

- Line 207: Abortion of ovules after double fertilization. On this topic, it is necessary to relate the different abortions with the pollination tests that led to fertilization.

- It is not clear from the text (Results) whether all ovules are fertilized in pollinated flowers. This needs to be made clearer. Furthermore, it is also necessary to check whether the development of unfertilized ovules from pollinated flowers is different from those of non-pollinated flowers.

Additional comments

- line 75: replace “because of” by “due to”

- In Figure 1C only one pistil is shown, so change the sentence in lines 127-128 to "The pistil of C. henryi comprise the stigma, style, and a large ovary (Fig. 1c)."

- line 172: check "Fig. 4gg, h"

- line 176: replace “type of the endosperm was nuclear endosperm” by “type of the endosperm was nuclear”

- line 230: “impede”

- line 248: C. henryi was not written in italics. Also check throughout the text.

---

## Round 0.3 · Minor Revisions

Thank you for submitting your revised version. I apologize for taking a bit of time to get the decision back to you.

One of the reviewers is fully satisfied. The other reviewer has a two notable concerns (in addition to various minor issues of wording that will be easy to fix).

The first concern is supplemental table 1, which contained the raw data. In the earlier round reviewers and me both found this table difficult to read. Your solution was to drop it. But that is not acceptable, you need to include raw data in the supplement. Instead, you have to work on making the table readable. Perhaps ask some of your colleagues to have a look and see if they can understand.

The second concern is with figure too. The images are far too small. The features are difficult to see and the annotation texts obscure the features of interest. For example, figure 3b is supposed to show a pollen tube breaking thru a synergid cell and releasing sperm. Another example: we are expected to see a nucleolus in figure 3p but in fact we can barely see the nucleus. None of this, or other things you mention, can be seen in the tiny images. Higher magnification images must be shown. One approach would be to break this into two figures, or even three, and double or more the area of each panel. Perhaps a better approach would be to couple these low-mag survey views with higher magnification images of the cells of interest. Perhaps a combination of those approaches would work.

As an aside, the panels in figure 4 and 5 are generally large enough although larger ones for figure 5 would be better.

As before, I also read your text carefully. Generally, your text is much improved. I found a smaller number of places where I edited for clarity or validity. In some places you confuse writing about what you observed with what is true. The first is a statement of fact and the second is an inference. I have uploaded a marked-up text. Please incorporate the changes shown, unless you have a specific reason not to do so. I included a few notes about minor issues; I will not repeat them here.

When you submit your revised version, include your responses to both reviewer 3 and to me in a single file (the response or rebuttal letter). Also include track changes for the both changes I have indicated in the marked-up text file when you use them and for anything that you introduce.

Feel free to contact me off-line if you have specific questions. I look forward to getting the revised text.

Reviewer 3 ·

Basic reporting

Basic Reporting: Basic reporting criteria are as before, now with additional concern that raw data is not shared (see comment in "Additional Comments" section).

Experimental design

Experimental Design: Experimental design is as before, with added clarification of experimental design.

Validity of the findings

Validity of Findings: Validity of finding criteria are as before, with continued concerns about the low resolution of microscopy images making it difficult to verify the stated findings. Raw data for Table 1 is no longer provided.

Additional comments

Line 140: I don’t understand the use of the term “buds” here. Would “structures” be more appropriate?

Line 143: Do you mean “polygonum-type” instead of “polygonal”? Also, this sentence makes it sound like the integuments and nucellus are part of the embryo sac (instead of part of the ovule).

Lines 152-166: As mentioned in previous rounds of review, the low resolution of the microscopy images in Figure 3 make it difficult to see the features (namely, sperm nuclei in 3b and 3c) that the authors describe. I appreciate that sperm nuclei are very small, which is why images of higher magnification and/or higher resolution are needed to clearly show these features in the figure.

Line 213: delete the redundant “, the dormancy period was very short, and”

Line 219: change to “develops into a seed”

Line 246-247: as asked in an earlier round of review, what is the evidence for the chestnut mother plant selecting highest quality ovules? It is shown that a subset of ovules are fertilized and that a subset of seeds develop to maturity, but the relative quality of the ovules was not assessed. The authors addressed this by changing the wording in another area of the paper (a few lines down in the conclusions), but it should also be changed here.

Table 1, Notes: Should “Date” be “Data”?

Supplementary Table 1: authors have deleted this supplementary table in response to request for clarification, but it is necessary to provide raw data.

·

Basic reporting

The revised version has been improved and reads well.

Experimental design

The methodology is better written in this new version and my doubts have been resolved.

Validity of the findings

To my questions the authors made the necessary adjustments or justified the results. The issue of pollination experiments, which could compromise the validity of the results, was improved. The discussion and conclusions are well stated.

Additional comments

The suggested changes have been adhered and I have no further comments.

---

## Round 0.4 · accepted · Accept

While I retain reservations about the quality of the micrographs in figure 3, I think that your statements in the paper probably reflect your observations. Readers who wish to challenge your interpretations are welcome to. Thank you for submitting to PeerJ.